

# Acute effects of lower trapezius activation exercises on shoulder muscle activation during overhead functional tasks in symptomatic and asymptomatic adults

Sophia Abiara[1], Vivian Heinrichs[1], Annaka Chorneyko[1] and
Angelica E. Lang[1,2]

[1] Canadian Centre for Rural and Agricultural Health, University of Saskatchewan, Saskatoon, SK, Canada
[2] Department of Medicine, University of Saskatchewan, Saskatoon, SK, Canada

Corresponding author
Angelica E. Lang,
angelica.lang@usask.ca

## ABSTRACT

**Introduction:** Shoulder musculoskeletal disorders are associated with atypical kinematics and muscle activity. Exercises to activate weakened muscles may reduce pain and disability. The objectives of this study were to test the effectiveness of exercises for activating the lower trapezius and to compare changes in shoulder muscle activity during functional tasks before and after the exercises in symptomatic and asymptomatic groups.

**Methods:** Eighteen adults, nine with subacromial pain syndrome and nine asymptomatic controls, participated in this study. A within-session repeated measures case-control design was employed. Participants performed overhead functional tasks before and after completing lower trapezius activation exercises. Electromyography (EMG) data of scapular muscles were captured during the tasks and exercises. One-way analysis of variance (ANOVA) compared muscle activations during the exercises. Paired t-test statistical parametric mapping assessed changes in muscle activity after the exercises.

**Results:** Lower trapezius activation was highest during the Trapezius Muscle Exercise for both groups. Post-exercise, a transient increase in lower trapezius activation was observed in the pain group during the Comb Hair task ($p = 0.0012$, $d = 1.55$) and the no pain group in the Overhead Reach task ($p < 0.001$, $d = 1.38$), but this effect did not persist in either group.

**Discussion:** The exercise protocol successfully increased lower trapezius activation immediately post-exercise, but the effects were short-lived. The findings suggest that while lower trapezius activation exercises can temporarily alter muscle activation, their efficacy for acute prevention or rehabilitation is limited. Further research is needed to explore the effects of longitudinal training programs on functional task performance.

## INTRODUCTION

Upper extremity musculoskeletal complaints are common (*Luime et al., 2004*; *Roquelaure et al., 2006*). Of those who present with musculoskeletal health issues, up to 26% an estimated 24% of those are dealing with shoulder pain (*Walker-Bone et al., 2004*) and the vast majority of those individuals have subacromial pain syndrome (SAPS) (*van der Windt et al., 1995*; *Virta, Brox & Eriksson, 2012*). SAPS includes conditions such as rotator cuff tendinitis, tendinosis, incomplete rotator cuff tears, biceps tendinitis, and bursitis, with diagnoses focused on symptoms rather than a single, clearly defined pathology (*Diercks et al., 2014*). SAPS is associated with abnormal scapular kinematics, and it is generally thought that the atypical muscle activation that occurs with SAPS may be a contributing factor in the development of the syndrome (*Chester et al., 2010*; *Diederichsen et al., 2009*; *Lin et al., 2006*; *San Juan et al., 2016*).

The lower trapezius portion of the trapezius muscle steadies the scapula and works in a couple with the upper trapezius to upwardly rotate and posteriorly tilt the scapula (*Cools et al., 2004*; *Ludewig, Cook & Nawoczenski, 1996*). Those who experience shoulder pain elicit different muscle activation patterns (*Mackay et al., 2023*; *Michener et al., 2016*), characterized by increased activation of the upper trapezius and decreased activation of the lower trapezius and serratus anterior (*Lin et al., 2006*; *Reijneveld et al., 2017*).

Specific exercises can activate the lower trapezius. Indeed, exercises that sufficiently activate the lower trapezius may improve clinical and biomechanical outcomes (*Garcia et al., 2023*; *Mendez-Rebolledo et al., 2022*; *Schory et al., 2016*). Prone exercises at low arm elevation (*Garcia et al., 2023*) or with external rotation (*Mendez-Rebolledo et al., 2024*) promote favourable lower trapezius activity over upper trapezius. Scapular exercise protocols with these types of exercise can improve lower trapezius activity, head and shoulder posture, pain, and strength over time (*Go & Lee, 2016*; *Kim, Chul Lee & Yoo, 2018*; *Park & Lee, 2020*). Additionally, acute sessions with biofeedback have also successfully modified muscle activation and scapular kinematics in healthy individuals for applications to injury prevention programs in workers (*Huang et al., 2013*; *San Juan et al., 2016*). However, these changes were documented in planar arm elevation which may not reflect muscle activity occurring during work or daily life.

There are movement or postural benefits from even acute training sessions, which may be of particular relevance for preventative ergonomic or work-fitness programs (*San Juan et al., 2016*). However, the acute effects of scapular exercises on shoulder musculature and kinematics during functional, work-related tasks, such as reaching and lifting, remain to be explored. Further, while biofeedback is an effective tool for activating desired muscles (*Antunes, Carnide & Matias, 2016*; *Mackay et al., 2023*; *Riek, Pfohl & Zajac, 2022*), this equipment may not be readily available in many rehabilitation or work environments. Indeed, for implementation of exercise training for acute effects in the workplace or in the home, the inclusion of any external equipment may not be feasible. Therefore, the efficacy of a simple, traditional exercise activation protocol with minimal equipment for application to functional movements needs to be examined in people with (symptomatic)

 

and without shoulder pain (asymptomatic) for potential preventative and treatment applications.

The objectives of this study are:

(1) To define how the trapezius muscles (upper trapezius (UT), middle trapezius (MT), lower trapezius (LT)) and serratus anterior (SA) activate during select scapular muscle exercises in individuals with and without shoulder pain

(2) To compare changes in shoulder muscle activity during overhead functional tasks before and after the acute exercises in both groups

It was hypothesized that the selection of lower trapezius-focused activation exercises would activate the lower trapezius muscle more than the upper trapezius in all participants and, following an acute bout of training, lower trapezius activation would increase during functional task performance in both groups.

## METHODS

### Participants

Two groups were recruited for this study: people with symptomatic subacromial pain syndrome and sex-matched healthy controls from a previous dataset. All data were collected in the Shoulder Health and Ergonomics Research Lab from May 2023 to April 2024. An *a priori* between-factors repeated measures ANOVA sample size calculation using an effect size of 0.83 (*San Juan et al., 2016*), power set to 0.8 (*Robinson, Vanrenterghem & Pataky, 2021*), and alpha set to 0.05 determined a minimum number of 16 participants were needed (*Faul et al., 2007*). A total of 18 participants were included in this study, divided evenly into the two groups. However, as the control group was originally recruited for a different protocol, the two groups were matched by sex and height, but not age.

Participants in both groups were adults over the age of 18. Participants in the pain group were confirmed to have shoulder pain for at least 3 months and screened with shoulder impingement tests (painful arc, Hawkins–Kennedy, infraspinatus, and empty can; *Lowry et al., 2021*; *Moen et al., 2010*). Participants in the control group were free from upper extremity musculoskeletal impairments. Exclusions criteria for both groups included history or presence of previous shoulder surgery, significant cardiac disease, heart rhythm disorders, pregnancy, inability to lift their arms past 90 degrees, and allergy to adhesives. The study protocol was approved by the University of Saskatchewan's research ethics board (Bio #3796).

### Procedures

This study employed a within-session repeated measures case-control design. Details are described below, but briefly, after sensor set up, each participant performed the functional tasks, followed by the selected activation exercises, and then repeated the same functional tasks.

**Table 1  Surface EMG electrode placement.**

| Muscle | Placement |
| --- | --- |
| Upper trapezius | Midway between the spine and the acromion on the upper back |
| Middle trapezius | Midway between the medial border of the scapula spine and T3 |
| Lower trapezius | On mass that arises when the participant retracts the scapula, midway between the scapular spine and T7, placed on an oblique angle |
| Serratus anterior | In axillary region, level with the inferior tip of the scapula, anterior to the latissimus dorsi muscle |

**Table 2  Maximum voluntary contraction positions.**

| Muscle | Position | Resistance |
| --- | --- | --- |
| Upper trapezius | Participant seated. Arm abducted to 90°, elbow bent to 90° and forearm parallel with the floor | Arm abducted and shrugged with resistance applied above the elbow |
| Lower trapezius | Participant prone. Arm abducted to 120° with thumb up and elbow straight, parallel to the floor | Arm lifted upward with resistance applied at the wrist |
| Middle trapezius | Participant seated. Arm abducted to 90°. Elbow bent to 90° with forearm parallel to the floor. | Arm pulled backward with resistance applied just above the elbow |
| Serratus anterior | Participant standing, slightly bent forward with torso curled anteriorly and scapulae protracted, with hands clasped | Push hands together and simultaneously pull elbows downward |

After providing informed written consent, participants completed the Quick Disability of Arm, Shoulder, and Hand (QuickDASH) and sensors for electromyography (EMG, Delsys Trigno™ Wireless EMG sensors; Delsys, Inc, Natick, MA, USA) were affixed to the upper body of the participants bilaterally on the UT, MT, LT, and SA as per previous published guidelines (*Criswell, 2010*) (Table 1). Before applying the sensors, the skin was prepped by shaving the area and cleansing with 70% isopropyl alcohol swabs. Participants completed maximum voluntary contractions (MVCs) in four positions to elicit the maximum activity in the targeted muscle for normalization (Table 2). To minimize potential fatigue effects, only one round of MVCs was performed. Participants were allowed at least 1 min of rest between exertions. EMG data were sampled at 2,000 Hz.

Motion data were also collected to identify relevant movement cycles for the EMG analysis. Reflective markers adhered to the skin or tight-fitting clothing (for torso markers) were used to track the torso and humeri movement. International Society of Biomechanics standards determined the placement of the markers and passive motion capture cameras (Vicon Motion Systems, Oxford, UK) tracked the movement. The marker position was synced with the EMG and sampled at 100 Hz.

The functional experimental tasks were performed twice: once before and once after the muscle activation exercises and in the same order each time. Overhead motions were the focus of this study as the trapezius muscles are primary contributors to scapular motion in arm elevation (*Ludewig, Cook & Nawoczenski, 1996*). First, participants performed the Comb Hair task. Holding a comb in their hand, they were instructed to bring the comb to their forehead and mimic the movement of combing their hair backwards. Next, for the Overhead Reach, participants were seated in front of a set of shelves with their hand resting on the shelf. They were instructed to move a 1 kg bottle from a lower shelf to a high shelf

set at 1.5 m off the ground. The final task was the Overhead Lift. Starting with both hands on the milk crate, participants moved the 8 kg weighted milk crate from a shelf at waist level to a higher shelf positioned at forehead level. Three repetitions were completed on each arm for unilateral tasks (Comb Hair, Overhead Reach), and three repetitions in total were for the bilateral task (Overhead Lift).

After the first round of the functional tasks, each participant completed the muscle activation exercises. The four lower trapezius activation exercises were taken from *Park & Lee (2020)* (Fig. 1). Each exercise was performed three times for 10 s at a time, with at least 1 min for rest allowed between each repetition to mitigate fatigue. This protocol was chosen not only based on the benefits reported (*Park & Lee, 2020*), but also the ease in performing the tasks for participants and potential future patients. No specialized equipment is required to perform these exercises. The first exercise was scapula setting (Exercise A: Fig. 1A). In this exercise, participants were seated and instructed to pull their chin into their neck and position their arms into a W with the palms facing forward. With their chest pulled forward, participants retracted their scapula. The next exercise was the Modified Prone Cobra (Exercise B: Fig. 1B). Participants lie prone on a massage table with their arms at their sides. Participants were instructed to lift their chest upward and pull their scapula downwards. The third exercise was the Wall Slide (Exercise C: Fig. 1C). Participants stood against the wall and began with their arm abducted to 90°, their elbows bent to 90°, and their palms facing forward. Participants were instructed to slowly raise their arms up and down while keeping their arms close to the wall. Stage 3 of the Trapezius Muscle Exercise progression was used as the fourth exercise (Exercise D: Fig. 1D). Participants lay prone on the massage table with their thumbs up, their arms abducted above 90°, and their elbow joint flexed. Participants were instructed to lift their chest upward and pull their scapula back and downwards.

Immediately following the activation exercises, all three functional tasks were repeated as above.

## Data analysis

Custom MATLAB codes were used to process the kinematic and EMG data. To eliminate heart rate artifacts, the EMG data were high pass filtered at 30 Hz (*Drake & Callaghan, 2006*). Using a cutoff of 3 Hz, a second order single pass Butterworth Filter was used to linearly envelope the data (*Waite, Brookham & Dickerson, 2010*). The linear enveloped signal was normalized to the maximum MVC value for each muscle. The UT/LT and UT/SA activation ratios were calculated for each trial from the normalized data. Values over 1 indicate higher relative upper trapezius activity. For each task, the time series normalized muscle activation for each relevant cycle was extracted for analysis. Movement cycles were defined from kinematic data as described below.

A low-pass, fourth-order, zero-lag Butterworth filter with a 6 Hz cut-off was used to filter the raw kinematic data (*Winter, 2009*). Humeral elevation was calculated as the angle between the long axes of the humeri and torso and was used to define movement cycles. The movement of the humerus five degrees marked the beginning of

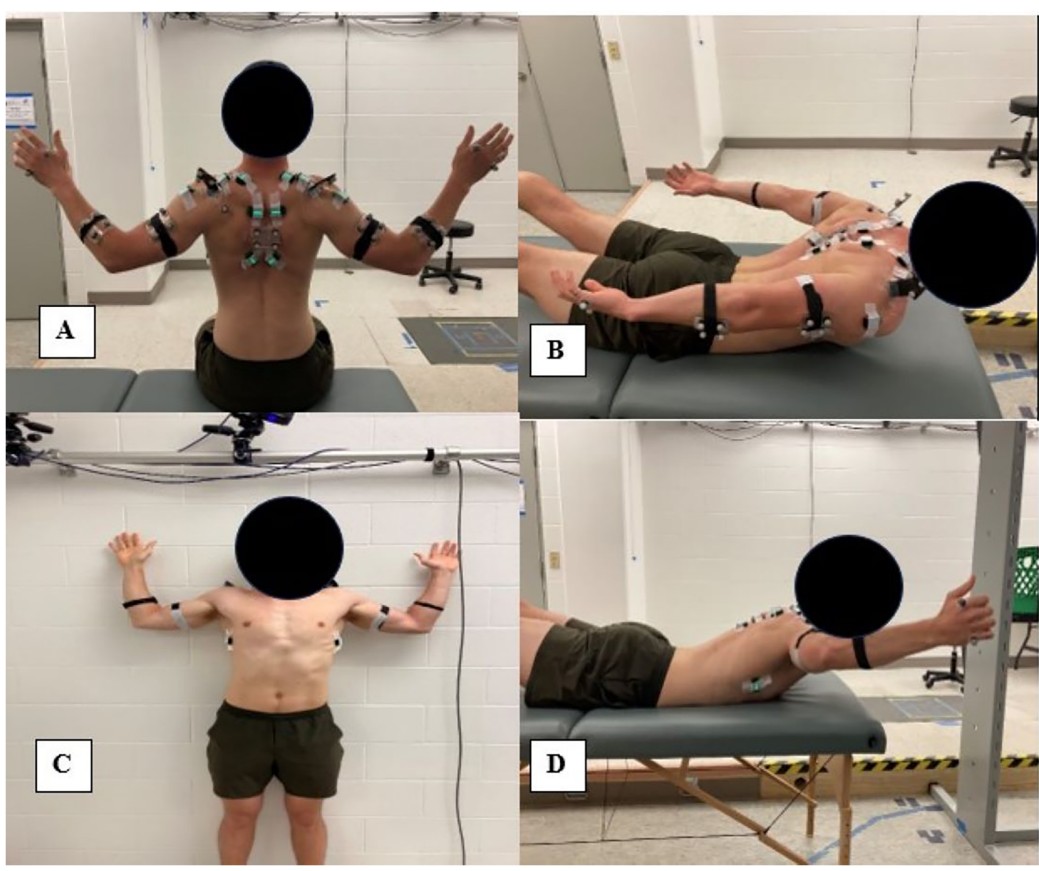

**Figure 1** Lower trapezius exercises. (A) Scapula Setting; (B) Modified Prone Cobra; (C) Wall Slide; (D) Trapezius Muscle Exercise.

each task (0%). Task completion was defined as when peak humeral position was attained (100%).

## Statistical analysis

For the statistical analysis, all shoulders from pain group participants meeting the inclusion criteria were included (15 shoulders), while all shoulders from the control group were included (18 shoulders). To compare muscle activity in the exercises, mean EMG for each muscle during the middle 30 s of each activation exercise was calculated and averaged as the dependent variable. A two-way mixed methods ANOVA ($p < 0.05$) was conducted to evaluate the effects of exercise type and group (pain $vs$ no pain) on muscle activation during the exercises using SPSS Statistics, version 27 (SPSS Inc., IMG, Chicago, IL, USA) and $post$-$hoc$ Tukey analyses ($p < 0.05$) identified group and type differences. Partial eta squared ($\eta_p^2$) was used to assess effect sizes of differences present during the exercises, with 0.01 indicating low effect size, 0.06 moderate effect, and 0.14 large effect (*Cohen, 1988*).

For the functional tasks, the change in muscle activity or activation ratio from pre- to post-exercises (positive change indicates a post-exercise increase) were explored with

**Table 3 Participant demographics (mean ± standard deviation unless otherwise noted).**

|  | Pain Group (n = 9) | Control (n = 9) | p-value |
|---|---|---|---|
| Sex (F/M) | 6/3 | 6/3 | – |
| Age (years) | 36.9 ± 14.6 | 23.9 ± 2.03 | 0.018 |
| Weight (kg) | 69.4 ± 14.7 | 72.0 ± 12.8 | 0.693 |
| Height (m) | 1.69 ± 0.08 | 1.69 ± 0.09 | 0.901 |
| Hand dominance | 9R | 9R | – |
| Pain onset (years) | 6.9 ± 6.7 | – |  |
| QuickDASH score (/100) | 27.0 ± 13.9 | 2.5 ± 5.3 | <0.001 |
| Injured shoulder | 1R/2L/6B | – | – |

**Note:**
R, right; L, left; B, both.

**Table 4 Mean (standard deviation) muscle activation (%MVC) during exercises for both groups.**

|  | Group | Exercise A | Exercise B | Exercise C | Exercise D | $\eta_p^2$ |
|---|---|---|---|---|---|---|
| UT | No pain | 7.0 (3.9) | 4.4 (2.1) | 12.1 (4.3)* | 19.2 (5.9)[†] | *0.234 |
|  | Pain | 5.7 (2.7) | 6.6 (5.1) | 24.3 (10.4)* | 26.7 (11.2)[†] | [†]0.724 |
| LT | No pain | 13.5 (8.1) | 15.3 (10.3) | 22.9 (17.4) | 35.6 (17.3)[†] | [†]0.487 |
|  | Pain | 14.0 (5.9) | 21.4 (8.4) | 22.4 (9.7) | 34.1 (19.6)[†] |  |
| MT | No pain | 29.1 (19.3) | 21.7 (12.4) | 18.2 (15.1) | 34.2 (19.1) | – |
|  | Pain | 19.9 (8.3) | 24.8 (11.9) | 25.0 (13.3) | 26.3 (9.5) |  |
| SA | No pain | 5.1 (3.8)[†] | 10.8 (7.0) | 15.5 (6.9) | 15.7 (9.8) | [†]0.432 |
|  | Pain | 5.0 (5.6)[†] | 13.2 (11.9) | 20.0 (11.2) | 19.0 (16.0) |  |
| UT/LT | No pain | 1.81 (1.43) | 0.59 (0.52)[†] | 1.32 (1.12) | 0.73 (0.39) | [†]0.356 |
|  | Pain | 1.52 (1.02) | 0.36 (0.24)[†] | 1.41 (0.57) | 1.21 (0.99) |  |

**Notes:**
Exercise A: "Scapula Setting", Exercise B: "Modified Prone Cobra", Exercise C: "Wall Slide", Exercise D: "Trapezius Muscle Exercise". UT, Upper Trapezius; LT, Lower trapezius; MT, Middle Trapezius; SA, Serratus Anterior.
* Indicates significant difference (<0.05) between groups for that muscle and exercise.
[†] Indicates significant difference between exercises within a muscle (row).

statistical parametric mapping (SPM) ($p < 0.05$) (*Kobayashi et al., 2022*; *Pataky, 2010*) in MATLAB. SPM was chosen for the analysis because it allows for exploration of all potential pre-post changes across the entire waveform. Cohen's d effect sizes assessed the maximum between-group differences for each functional task, with 0.2 indicated low effect size, 0.5 moderate effect, and 0.8 large effect (*Cohen, 1988*).

## RESULTS

Participants in the pain group were matched by sex and height from a previous younger (ages 18–35) dataset that underwent the same protocol (Table 3). The control group was younger and reported lower QuickDASH scores.

Muscle activity varied in the exercises in both groups. Lower trapezius activation was highest in both groups for Exercise D ($p < 0.001$, partial eta squared ($\eta_p^2$) = 0.478) (Table 4).

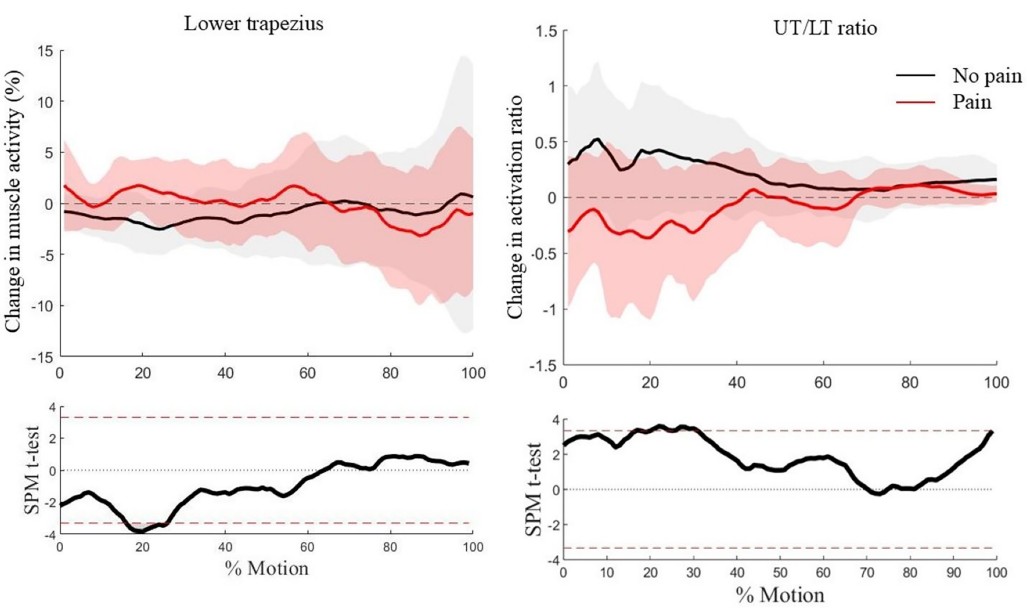

**Figure 2 Waveform analysis of the pre- to post-exercise change in lower trapezius activation (left) and change in upper to lower trapezius (UT/LT) activation ratio (right) ($p < 0.05$) during the comb hair task.** Average waveforms (solid line = mean, shaded areas = one standard deviation) are on the top and SPM findings on the bottom.

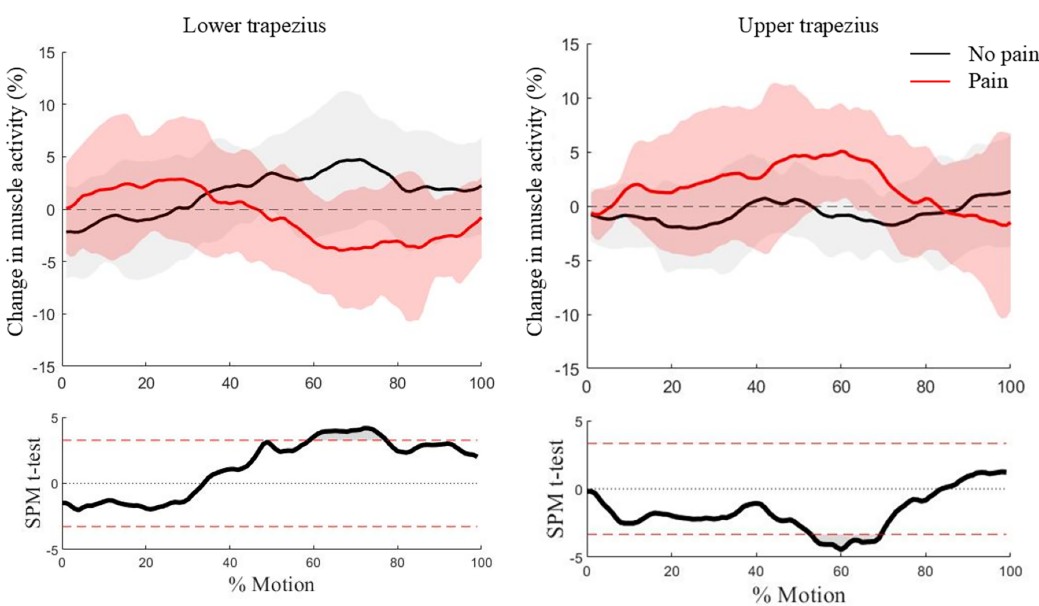

**Figure 3 Waveform analysis of the pre- to post-exercise change in lower trapezius activation (left) and change in upper trapezius activation (right) ($p < 0.05$) during the overhead reach task.** Average waveforms (solid line = mean, shaded areas = one standard deviation) are on the top and SPM findings on the bottom.

Upper trapezius activation was also greatest in Exercise D for both groups ($p < 0.001$, $\eta_p^2 = 0.724$) (Table 4). For Exercise C, there was a significant difference between groups for the upper trapezius ($p = 0.021$, $\eta_p^2 = 0.234$), with the pain group eliciting higher upper trapezius activation (Table 4). The upper trapezius to lower trapezius activation ratio was the lowest for both groups in Exercise B ($p < 0.001$, $\eta_p^2 = 0.356$, Table 4).

During the functional tasks, some significant differences were observed in the individual muscle activation of the upper and lower trapezius and the upper to lower trapezius activation ratio. Lower trapezius activation increased post-exercise for the pain group relative to the no pain group at approximately 20% of the motion in the Comb Hair task ($p = 0.0012$, maximum difference = 3.4%, Cohen's $d$ =1.55, Fig. 2). The UT/LT ratio also decreased post exercise in the pain group in this task ($p < 0.001$, maximum difference = 0.8, $d = 1.16$, Fig. 2). However, these changes did not continue into the proceeding tasks. In the Overhead Reach, lower trapezius activation in the no pain group was greater post-exercise relative to the pain group ($p < 0.001$, maximum difference = 8.4%, $d = 1.38$, Fig. 2). Additionally, upper trapezius activation increased in the pain group post-exercise ($p < 0.001$, maximum difference = 5.9%, $d = 1.55$, Fig. 3) in Overhead Reach. No differences existed during the Overhead Lift, the final task.

## DISCUSSION

The present study aimed to characterize the effectiveness of select LT activation exercises at recruiting the LT preferentially over the UT and to determine if those exercises had acute effects on muscle activity during a functional task protocol in individuals with and without shoulder pain. Some exercises were successful at preferentially activating the LT over the UT, but overall the exercises did not activate the LT to the desired relative activation. Post-exercise, LT activation increased in both groups, but in different tasks.

The first hypothesis, positing that the exercises would effectively activate the LT preferentially over the UT for both groups, was partially supported, as this occurred in two of the four exercises. LT activation was higher than UT activation for the Modified Prone Cobra (Exercise B) for both groups and in the Trapezius Muscle Exercise (Exercise D) for the no pain group but not for the pain group. According to the literature, ratios lower than 1.0 for the UT/LT ratio are preferred (suggesting the LT is more active than the UT) (*Garcia et al., 2023*), although lower than 0.6 are ideal (*Mendez-Rebolledo et al., 2021*). While the direct connection between specific UT/LT ratio values and dysfunction remains elusive, ratios >1.0 are considered non-optimal for rehabilitation interventions (*Cools et al., 2007*), as they promote greater UT activation, which is often already present in pain groups and considered to be a negative adaptation (*Ludewig & Cook, 2000*). The ratio during Modified Prone Cobra was below the 0.6 threshold for both groups, while the mean ratios for the Wall Slide and the Scapula Setting were both over 1.0, suggesting that the UT was relatively active in both exercises. In the Trapezius Muscle Exercise, the no pain group ratio was below 1.0, but the pain group ratio was above this threshold. Therefore, the exercises as a set were not successful in activating the LT relative to the UT in the pain group, and only the Modified Prone Cobra (Exercise B) can be recommended. These findings indicate that exercise choices may be insufficient to elicit preferred UT/LT ratios

in both injured and control populations. Other, similar exercises with lower arm elevation may elicit the desired LT ratio activation (*Garcia et al., 2023*) in symptomatic and asymptomatic groups.

The second hypothesis, which stated that lower trapezius activity would change after activation exercises, was also partially supported. During the first task, Comb Hair, lower trapezius activity increased at the beginning of the movement cycle for the pain group, which is expected to have compromised LT activation (*Michener et al., 2016*). Additionally, LT activation increased in the no pain group in the Overhead Reach. The LT activation increase is the desired adaptation. However, this change did not persist beyond the Comb Hair task for the pain group, as LT activation decreased and UT activation increased in the Overhead Reach in this group. No changes existed for either group in the Overhead Lift (last task). These findings suggest that the exercise protocol may have successfully increased the LT in the pain group in the Comb Hair task, but not other overhead functional movements. Conversely, another explanation for the presence of changes in the Comb Hair only could be the order of the tasks. As the Comb Hair task was always performed first, the exercise protocol may have instead induced a transient increase in LT but then the effects washed out, similar to previous health-related acute interventions (*Andersen et al., 2013*; *Chang et al., 2025*). Short-lived effects could be due to insufficient motor learning or persisting pain, or a combination of both, overriding the acute neuromuscular effects; however, further research is needed to elucidate if the changes were task or order specific.

It is unclear why the pain group demonstrated improved muscle activation in the first task, Comb Hair, only, while the no pain group improved in the Overhead Reach only. Differences with large effect sizes existed between groups in these tasks. As the pain group is expected to have an impaired UT/LT ratio, it is possible the exercises were initially more effective for them. However, the difference between the groups in Overhead Reach may be due to the demands of the task: the Overhead Reach has been shown to elicit impaired scapular movement in pathological groups (*Lang et al., 2022*, *2019*; *Michener et al., 2016*; *Spinelli et al., 2016*). Therefore, post-exercise, the no pain group with unimpaired activation may have been positively affected by the training in this task while the pain group was not, but more work is needed to confirm why this occurred. As scapular muscle activity and recruitment are influenced by task characteristics and performance, such as movement velocity (*Mendez-Rebolledo et al., 2019*, *2018*), the differing task demands, such as load, arm position, and participant chosen movement speed, may contribute to the underlying mechanisms for muscle activation differences. Regardless, positive compensations in either group did not remain throughout the Overhead Lift. Further research is needed to assess if positive exercise effects are task dependent, and if so, how to leverage changes in all overhead motions.

Some considerations delimit the interpretation of these findings. The two study groups, while matched by sex, were not matched by age, which has been shown to affect shoulder motion (*Kwon et al., 2021*). As a result, the pain group was significantly older than the no pain group. However, research has also indicated that for groups under 65, age does not influence motion in the three tasks tested (*Waslen, Friesen & Lang, 2023*). Due to the

length of the study procedures, only one round of MVCs was performed, which may affect normalization. The effect of the exercises was tested in an acute training session after only completing three-ten second repetitions of each exercise. While this was the procedure from previous research (*Park & Lee, 2020*), it may have been insufficient to elicit true, lasting effects. With regards to the functional tasks assessed, future research should consider randomizing the task order to determine if timing and order of the tasks or the specific tasks and movement demands are responsible for the differing changes by group after the exercises. The level of pain experienced during task performance was also not considered in the analysis and should be considered in future research. Finally, while this cohort met our inclusion criteria, based on previous, similar research (*Lawrence et al., 2014*; *McClure, Michener & Karduna, 2006*; *Michener et al., 2016*; *Turgut, Duzgun & Baltaci, 2017*), this group did not present with compromised LT activation (*Lang & Kim, 2025*), although this was not explicitly assessed in the current study. Future work should consider testing individuals with compromised LT activation to better understand the potential for this exercise protocol. While these findings represent a potentially interesting application for short term alterations if exercise selection is improved, the effects of a longitudinal training program with or without alternative approaches, like manual cuing (*De Mey et al., 2013*), mirrors (*Louw et al., 2017*), or biofeedback (*Antunes, Carnide & Matias, 2016*; *Mackay et al., 2023*; *Riek, Pfohl & Zajac, 2022*) on functional task performance should be tested to understand the lasting preventative or therapeutic effects on people with shoulder pain.

## CONCLUSIONS

An acute exercise session of simple, lower trapezius focused exercises induced a small, increase in lower trapezius activation during functional movements in a pain group and a non-pain group. However, the selected exercises did not all achieve the desired activation, and the post-exercise effects did not last, with neither group demonstrating increased lower trapezius activation throughout all of the overhead functional movements. The current exercises may not be sufficient for acute rehabilitation or prevention of SAPS. Regardless, the initial positive changes suggest that an acute session of activation exercises is a potential avenue for continued exploration for altering muscle activation that could influence shoulder movement and subsequent injury-related biomechanics. This study is explored changes to muscle activity in functional tasks from a simple acute training session and can be used to inform further testing with asymptomatic and symptomatic populations.

### Funding
This work was supported the University of Saskatchewan's College of Medicine. The funders had no role in study design, data collection and analysis, decision to publish, or preparation of the manuscript.

## Grant Disclosures

The following grant information was disclosed by the authors:
University of Saskatchewan's College of Medicine.

## Competing Interests

The authors declare that they have no competing interests.

## Author Contributions

- Sophia Abiara performed the experiments, analyzed the data, prepared figures and/or tables, authored or reviewed drafts of the article, and approved the final draft.
- Vivian Heinrichs performed the experiments, analyzed the data, authored or reviewed drafts of the article, and approved the final draft.
- Annaka Chorneyko performed the experiments, analyzed the data, authored or reviewed drafts of the article, and approved the final draft.
- Angelica E. Lang conceived and designed the experiments, performed the experiments, analyzed the data, prepared figures and/or tables, authored or reviewed drafts of the article, and approved the final draft.

## Human Ethics

The following information was supplied relating to ethical approvals (*i.e.*, approving body and any reference numbers):

The University of Saskatchewan (Bio #3796) approved all study procedures.

## Data Availability

Raw data is available in the Supplemental Files.

## Supplemental Information

Supplemental information for this article can be found online at http://dx.doi.org/10.7717/peerj.19861#supplemental-information.

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
