# Peer review of "Acute effects of lower trapezius activation exercises on shoulder muscle activation during overhead functional tasks in symptomatic and asymptomatic adults"

_PeerJ, doi:10.7717/peerj.19861_

## Round 0.1 · original submission · Major Revisions

Thank you for your submission. Based on the reviewers’ comments, I kindly request that you address the following key issues to improve the clarity and scientific rigor of the manuscript: (i) please clarify methodological aspects, including the number of participants, inclusion and exclusion criteria, and whether pain was monitored during task performance; (ii) ensure greater precision in the interpretation of results, specifically, avoid drawing conclusions not fully supported by the study’s quasi-experimental repeated-measures design, which compares groups with and without pain but lacks random assignment; (iii) expand on the clinical interpretation of the UT/LT ratio and report effect magnitudes using numerical values; (iv) elaborate on the underlying mechanisms behind the observed differences in muscle activation across exercises. Prior studies have shown that factors such as fatigue, velocity, and task characteristics influence scapular muscle recruitment patterns and timing (Mendez-Rebolledo et al., 2016; 2018; 2019). Including this perspective would enhance the depth of the discussion and clarify the clinical relevance of the findings; (v) temper the conclusions in light of the study’s limitations; and (vi) update the state of the art by incorporating recent literature (2020–2025) on therapeutic exercise for scapular stabilizer muscles. The current references are largely outdated, limiting the manuscript’s ability to reflect recent advancements and potentially weakening the rationale for the study. Relevant examples include:

Comparative electromyographic study of scapular stabilizing muscles during five main rehabilitation exercises. doi:10.1097/PHM.0000000000002394

Electromyographic analysis of the serratus anterior and upper trapezius in closed kinetic chain exercises performed on different unstable support surfaces: a systematic review and meta-analysis. doi:10.7717/peerj.13589

Shoulder kinematics and muscle synergy during multi-plane humeral elevation and lowering. doi:10.1016/j.jbiomech.2025.112735

Scapular muscle activation at different shoulder abduction angles during Pilates Reformer arm work exercise. doi:10.3390/medicina61040645

Muscle coactivation changes following a fatiguing overhead drilling task: implications for subacromial impingement syndrome. doi:10.1016/j.apergo.2025.104470

Kinetic chain modifies muscle activation in adults with shoulder pain: a randomized cross-over trial. doi:10.1016/j.jse.2024.10.023

Reviewer 1 ·

Basic reporting

SAPS (Subacromial Pain Syndrome) is described as a syndrome encompassing a range of conditions, such as rotator cuff tendinopathy, subacromial bursitis, and others. It would be helpful to clarify that SAPS is more accurately considered a symptom-based syndrome rather than a single, clearly defined pathology. There remains some controversy in the literature regarding its diagnostic utility.

Although the aims of the study are clearly stated, there is currently no information provided about the number of participants, inclusion and exclusion criteria, or potential age and sex differences. While these details may be elaborated on later in the methods section, it would be beneficial to include a brief preview at this stage—for example: “This study will recruit N individuals from X groups…”
Regarding the phrase “functional, work-related tasks,” it would be preferable to define or provide examples of such tasks (e.g., overhead lifting, shelf placement), as the term “functional” may be interpreted rather broadly.

The authors rightly note the limited accessibility of biofeedback; however, no alternative approaches are suggested. It would be constructive to propose more accessible options, such as manual correction, the use of mirrors, or mobile applications with EMG-based feedback.
In “outcomes (14,15).” – a space should be added between the word and the citation.

Experimental design

Lines 84–87 – the numerical estimations presented are questionable. It may be more appropriate to refer to established guidelines for research in the stomatognathic system, as it is the anatomically closest system to the muscles examined in this study. Please consider citing the study [10.12659/MSM.948365] and noting that the sample size used in that research allowed only the detection of large effects at a power of 60%. If the authors are using different guidelines, this should be justified; however, in its current form, the rationale remains unclear.

Line 104 – “with isopropyl alcohol.” Please specify the percentage concentration of alcohol used.
Statistical Analysis – It would enhance the clarity of the statistical analysis to include a description of effect size thresholds.

Validity of the findings

Discussion – The statement “The first hypothesis was partially supported” should be expanded: What exactly was not supported and why? For example, does this refer to the absence of effect in the pain group or insufficient differences between the exercises?
Although the discussion references differences between groups and observed effects, no information is provided regarding p-values, confidence intervals, or effect sizes, which limits the reliability of the conclusions.

The interpretation of the UT/LT ratio would benefit from a broader clinical context. While a threshold value (e.g., 0.6) is mentioned, it is unclear what the clinical implications are. For instance, does a ratio of 1.0 represent a meaningful risk of dysfunction? It would be helpful to clarify the functional consequences.

The authors note that the improvement in LT activation in the pain group did not persist. This point should be further elaborated upon. For example, was the effect short-lived due to muscle fatigue, lack of motor learning consolidation, or persisting pain?

Mentions of LT activation during the “Comb Hair” or “Overhead Reach” tasks are not supported by numerical data—readers are left unaware of the magnitude of these changes. It may be helpful to incorporate a brief table or at least provide percentage values.

“Large effect sizes” are mentioned, but not developed further. If between-group differences yielded large effects, these should be reported explicitly (e.g., Cohen’s d), with reference to which exercises or tasks they were observed in.

Study limitations are discussed, but remain somewhat general. While the mention of age mismatch and limitations of MVC is appreciated, it would strengthen the discussion to consider additional constraints such as the absence of randomisation, the low number of exercise repetitions (3 × 10 s), and the lack of control for pain intensity during testing.

Need for further research – The conclusion could be improved by suggesting more specific directions for future studies. For example, should future research involve long-term intervention programmes? Should exercises with and without biofeedback be compared? Are there other relevant patient populations that should be investigated?

Finally, the conclusions should be slightly tempered, bearing in mind that this was a small-sample study.

Reviewer 2 ·

Basic reporting

Line 22 – Effectiveness for which goal?
Line 43 – Reference 2 does not apply to this statement
Line 45 – Reference 3 does not apply to this statement
Line 50 – The statement ‘The trapezius muscles are important for healthy scapular motion’ is vague and does not provide any particular information. I suggest deleting.
Line 74 – Objective number 2 is clearer in the abstract than it is here. This statement only references the post and uses the word ‘altered’, but it’s unclear whether you are looking at change or not, and altered with respect to what?
Line 78 – Based on the background provided that patients with SAPS have decreased activation of the LT, the hypothesis that both groups will increase activation is surprising.
Line 260 - cannot

Experimental design

Everything is clear and explained in enough detail. The only additional information I would like to see is whether the authors collected any pain levels during the tasks for the participants with pain and, if so, whether this was controlled as a potential covariate.

Validity of the findings

Line 208 – I don’t think you can make the claim ‘Post-exercise, a transient increase in LT activation existed, but the change was washed out or reversed in later tasks’. To be able to know this for sure, you would need to have the tasks in random order or some sort of standardized procedure to rule out that your exercise intervention wasn’t particularly effective at inducing changes in that particular task.
Line 224 – For the statement ‘During the first task, Comb Hair, lower trapezius activity increased at the beginning of the movement cycle for the pain group, which is expected to have compromised LT activation,’ it should be clarified that your cohort did not have compromised LT activation.
Line 227 – compensation? I suggest ‘adaptation’.
Line 228 – alteration? I suggest ‘change’
Line 228 – Same comment as before. I don’t think you can say it did not persist beyond… You should say which tasks showed changes and which ones didn’t. You can’t imply that the effect was washed out. This has not been tested. You can hypothesize and make clear that it’s your interpretation of what may have happened.
Line 235 – You can’t say ‘it quickly reversed’ for the same reason.
Line 246 – You may want to add here that the age difference between your groups was statistically significant.
Line 257 – I don’t think you can say transient for the same reasons above.

Additional comments

This is overall a good study that deserves to be published. It adds new information to the literature that is worth having out there. It's not very groundbreaking but follows a logical next step in what the literature needs in this field. I suggest minor changes to address my concerns, especially regarding the interpretation of the 'transient' changes.

---

## Round 0.2 · accepted · Accept

We thank the authors for submitting the revised version of the manuscript. After evaluating the changes made and considering that the authors have adequately addressed the main reviewers’ comments, including methodological clarification, refinement of result interpretation, expanded discussion on the UT/LT ratio, deeper analysis of neuromuscular mechanisms, more balanced conclusions, and updated literature, we are pleased to inform you that the manuscript has been accepted for publication.

Reviewer 1 ·

Basic reporting

This is acceptable

Experimental design

This is acceptable

Validity of the findings

This is acceptable

Additional comments

Congratulations to the authors.

Reviewer 2 ·

Basic reporting

Issues were resolved

Experimental design

Issues were resolved

Validity of the findings

Issues were resolved

Additional comments

Issues were resolved